# Brief communication: How deep is the snow on Mount Everest?

Wei Yang[1], Huabiao Zhao[1], Baiqing Xu[1], Jiule Li[1], Weicai Wang[1], Guangjian Wu[1], Zhongyan Wang[1], Tandong Yao[1]

[1]State Key Laboratory of Tibetan Plateau Earth System, Environment and Resources (TPESER), Institute of Tibetan Plateau Research, Chinese Academy of Sciences, Beijing 100101, China

*Correspondence to*: Huabiao Zhao (zhaohb@itpcas.ac.cn)

**Abstract.** Exploring the snow depth on Mount Everest has long been a topic of interest in this inaccessible place on our planet. Previously reported snow depths have been inconsistent and have large uncertainties. Here, we report the ground-penetrating radar survey of snow depth along the northern slope of Mount Everest in May 2022. Our radar measurements display a gradual increasing transition of snow depth along the north slope, and the mean depth estimates at the summit are 9.5±1.2 m. This updated snow depth on Mount Everest is much deeper than previously reported values (0.9-3.5 m).

## 1 Introduction

Mount Everest, one of the most inaccessible places on our planet, is considered to be the most iconic peak (Kang et al., 2022; Matthews et al., 2020). There are very strong scientific and public motivations for determining the snow depth on Mount Everest. Although China and Nepal jointly declared that the snow height of Mount Everest was 8848.86 metres above sea level (m asl) in 2020, the true rock height has not been precisely determined due to the unknown snow depth below. The snow depths at extremely high elevations may vary dynamically with different seasons and years. Knowledge about snow depths during different periods will be helpful for explaining the discrepancies in the reported snow heights on Mount Everest, which have been introduced by repeated surveys (Angus-Leppan, 1982; Chen et al., 2010; Xie et al., 2021). In addition, snow layering at mountain summits contains information about local seasonal snow accumulation and climate history. However, snow and ice recession is accelerating in almost all regions on Earth (Hugonnet et al., 2021; Kraaijenbrink et al., 2021). Similar to other snow/glacier-covered summits (Thompson et al., 2009), the snow and glaciers on Mount Everest are the sentinels for climate change and therefore offer a potential natural platform for understanding ongoing climate change at such extremely high elevations (Matthews et al., 2020; Potocki et al., 2022) and their possible widespread influence on the Asian Water Tower (Immerzeel et al., 2020). Comparisons of snow depth/stratigraphy during different periods may be potentially helpful for understanding the possible influence of anthropogenic climate change at extremely high elevations in the Himalayas (Brun et al., 2022; Pepin et al., 2022; Potocki et al., 2022).

Previously reported snow depths derived by different methods and instruments ranged from 0.92 m to 3.5 m on Mount Everest. In 1975, a Chinese expedition team reported an estimated snow depth of 0.92 m by inserting a wooden stake into the

snow (Chen et al., 2010). In 1992, a joint Chinese-Italian expedition team estimated a thickness of 2.52 m by inserting a steel stake into the snow (Chen et al., 2010). These results derived by stake methods were subjected to many factors such as snow density, stake length, and manpower issues at such harsh altitudes. Radio echo sounding is a suitable technique for imaging snow-ice environments and their internal structures (Rignot et al., 2013). In 2005, a Chinese mountaineering and surveying team claimed a snow depth of ~3.5 m by utilizing ground penetrating radar; however, the reported boundary between the snow and rock on the radar image was too ambiguous to provide an undisputed depth (Sun et al., 2006). In 2019 and 2020, various Nepalese and Chinese expedition teams measured the snow depth using different radar instruments; however, no results were reported. Supported by the Second Tibetan Plateau Scientific Expedition and Research, we organized "The Earth Summit Missions 2022" expedition during the period from April to May 2022. One of our key goals was to measure the snow depth on Mount Everest.

## 2 Data and method

Ground penetrating radar (GPR) is a powerful tool in the field of cryosphere that has been widely used to survey snow depth (Holbrook et al., 2016; Yamamoto et al., 2004). To maximize portability on Mount Everest, we conducted our GPR survey with a single transmitter–receiver antenna at a frequency of 1000 MHz using a Sensor & Software Pulse EKKO Pro system on May 4[th] 2022. In contrast with the previous radar survey conducted at the summit (Sun et al., 2006), our measurement started from the exposed metamorphosed limestone at an elevation that was approximately 15 m lower than the summit to ensure a gradual transition in the radar reflection profile and thus produce more easily post-discerning boundaries between the snow and rock (Fig. 1a). For all GPR measurement points, a portable global navigation satellite system (UniStrong G138BD) continuously recorded the antenna locations. We obtained a total of 57 radar wavelet traces at irregularly spaced intervals (~0.5-1 m along the north slope and 0.2-0.4 m at the summit) during 12:30-13:00 (Fig. 1b, Supplement Table S1).

The transmission velocity is the most critical parameter for estimating snowpack thickness. Because of the limited measurement time window in the so-called 'death zone', we did not measure common midpoint data to evaluate the transmission velocity of radar waves inside the snowpack on Mount Everest. In general, the transmission velocity in snow ranges from 0.20 m/ns to 0.27 m/ns, which depends on the snow properties (Kovacs et al., 1995; Fortin and Fortier, 2001; Singh et al., 2017). A transmission velocity of 0.23 m/ns was obtained in a snowpack according to radar measurements with a steel stake (40 cm in length and 2 cm in diameter) that was buried in the snowpack at an elevation of 7028 m in 2005 (Sun et al., 2006). Therefore, we adopted a mean transmission velocity of 0.23 m/ns in this study.

To produce radar images that were more suitable for straightforward interpretations, the raw GPR data were processed using the Sensors & Software EKKO_project processing package by applying a frequency bandpassing filter and time-variable gain corrections. The processing steps increased the signal-to-noise ratio to improve the imaging results while maintaining the original data signature, thus producing data that can be easily interpreted. The boundary between the snow and rock and the subsurface stratigraphies were visually traced.

## 3 Results and discussion

The radar wavelet traces showed strong signal contrast between the snow and the rock surface (the blue dashed curve in Fig. 1c). It displays a well-defined gradual trend of radar reflection along the direction from the exposed limestone to the summit (from wavelet No.1 to wavelet No.31), which indicates the thickening inclination of snow depth along the northern slope of Mount Everest. Such a thickening pattern agrees with the observed thick snowpack exposed by the nearby cliff and the topographic conditions for snow accumulation (Fig. 1a). It should be noted that such a measurement along the north slope was used only for the purpose of generating the post-discerning radar boundaries, and the measurement process could give different results if the measurement profile were moved a few metres to either side. The radar wavelet traces of the other 26 measuring points (Nos. 32-57), which are mainly concentrated at the summit (Fig. 1b), displayed similar radar reflections. Such homogeneity not only indicates the reliability of repeated radar measurements within this limited area, but also provide insights into the relatively flat topography along the ridge of Mount Everest.

The magnitude of the estimated snow depth on Mount Everest greatly depends on the choice of the mean transmission velocity. Taking the mean snow transmission velocity of 0.23 m/ns obtained at 7028 m asl on Mount Everest (Sun et al., 2006), we obtained the snow depth distribution from the starting measurement point to the summit (Fig. 1c). The maximum two-way travel time of the reflecting horizon of the rock surface was approximately 88 ns on Mount Everest. The snow depth estimates gradually increased from ~2.0 m near the start of the exposed limestone to a maximum of ~10.1 m along the north slope. The snow depth of a total of 26 measuring points concentrated at the summit was averaged to be approximately 9.5 m. Such thick accumulated snowpack on Mount Everest may be partially explained by the westerly-introduced snowfall accumulation on the eastern leeward side. Moreover, compared with the lower amount of snow accumulation along the unfavourable steep slope, our radar measurements covered the relatively flat platform on Mount Everest, which may provide favourable topography for snow accumulation.

Although the adopted transmission velocity in snow was determined at elevations of 7028 m on Mount Everest, some uncertainties may still be introduced by the distinct snow conditions at the summit (*e.g.* the snow density and snow properties). The colder air temperature and stronger wind levels at higher elevations may favour significant morphological changes, and thus the snowpack was compacted, resulting in high snow density. Therefore, if a higher mean snow density of ~500 kg/m$^3$ were assumed on Mount Everest, the mean transmission velocity would decrease to ~0.21 m/ns (Fortin and Fortier, 2001). The mean snow depth on Mount Everest would slightly decrease from ~9.5 m to ~8.7 m. The transmission velocity in snow generally ranges from 0.20 m/ns to 0.27 m/ns (Kovacs et al., 1995; Fortin and Fortier, 2001; Singh et al., 2017). Taking 0.20 m/ns and 0.26 m/ns as the possible lower and upper boundaries for uncertainty estimation, the mean depth estimates at the summit were 9.5 ±1.2 m in May 2022.

In fact, the snow depth on Mount Everest should display interannual variability because of the influences of snow accumulation and snow drift. According to the recall of mountaineers who reached the summit in 2021 and 2022, the previously exposed rock surface in May 2021 was covered by a snowpack of approximately 60-70 cm depth in May 2022. Our reported

snow depth for Mount Everest in 2022 is considerably deeper than the values that were previously reported during the past five decades (0.9-3.5 m). There is still a lack of evidence that the snowpack has become thicker or thinner in recent decades. Future repeated radar measurements at the summit would be helpful for evidencing such dynamic changes under climate change.

In addition to revealing the magnitude of the snow depth on Mount Everest in May 2022, the radar wavelet traces showed two possible subsurface reflections within the snowpack (the yellow dashed lines in Fig. 1c). The upper weak subsurface reflection displays a shallow trend from a burial depth of ~2-3 m along the north slope to ~0.8-1.0 m. Another weak reflection layer existed at a relatively uniform depth of approximately 4.5 m (Fig.1c). Such features may be attributed to the transition boundaries between fresh snow, compacted older snow and granular firn. However, this remains speculative due to the weak

signal contrast between layers.

## 4 Conclusions

      Overall, our measurements acquired in May 2022 provide the first clear radar image of the snowpack at the summit of Mount Everest. This updated snow depth on Mount Everest is considerably deeper than the values that were previously reported during the past five decades (0.9-3.5 m). Such efforts provide new insights for deciphering the true rock height and bottom

geomorphology of Mount Everest. It is worth noting that recent debates took place on the surface melting that occurred at extremely high elevations (above 8000 m asl) on Mount Everest (Brun et al., 2022; Potocki et al., 2022). Indeed, future snow core drilling and repeated ground penetrating radar measurements on Mount Everest are also necessary to not only increase our understanding of dynamic snow changes, but also to detect the possible influence of unprecedented anthropogenic climate change by exploring the snow stratigraphy and snowpack properties at the Earth's summit.

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

**Data availability**

The GPR data (.sgy) are available from the corresponding author on reasonable request.

## Author Contributions

T.Y. designed research; T.Y., H.Z. and W.Y. analyzed data; and T.Y., H.Z., W.Y., B.X., J.Li., W.W., G.Wu and Z.W. jointly discuss the result and wrote the paper.

## Acknowledgements

The study is supported by the Second Tibetan Plateau Scientific Expedition and Research Program (STEP) (2019QZKK0201), the National Natural Science Foundation of China (41961134035). We would like to sincerely acknowledge and express our deep appreciation to the editor Chris Derksen and two anonymous reviewers for their thorough review of this work.

## Competing interests

The authors declare no competing financial interests.

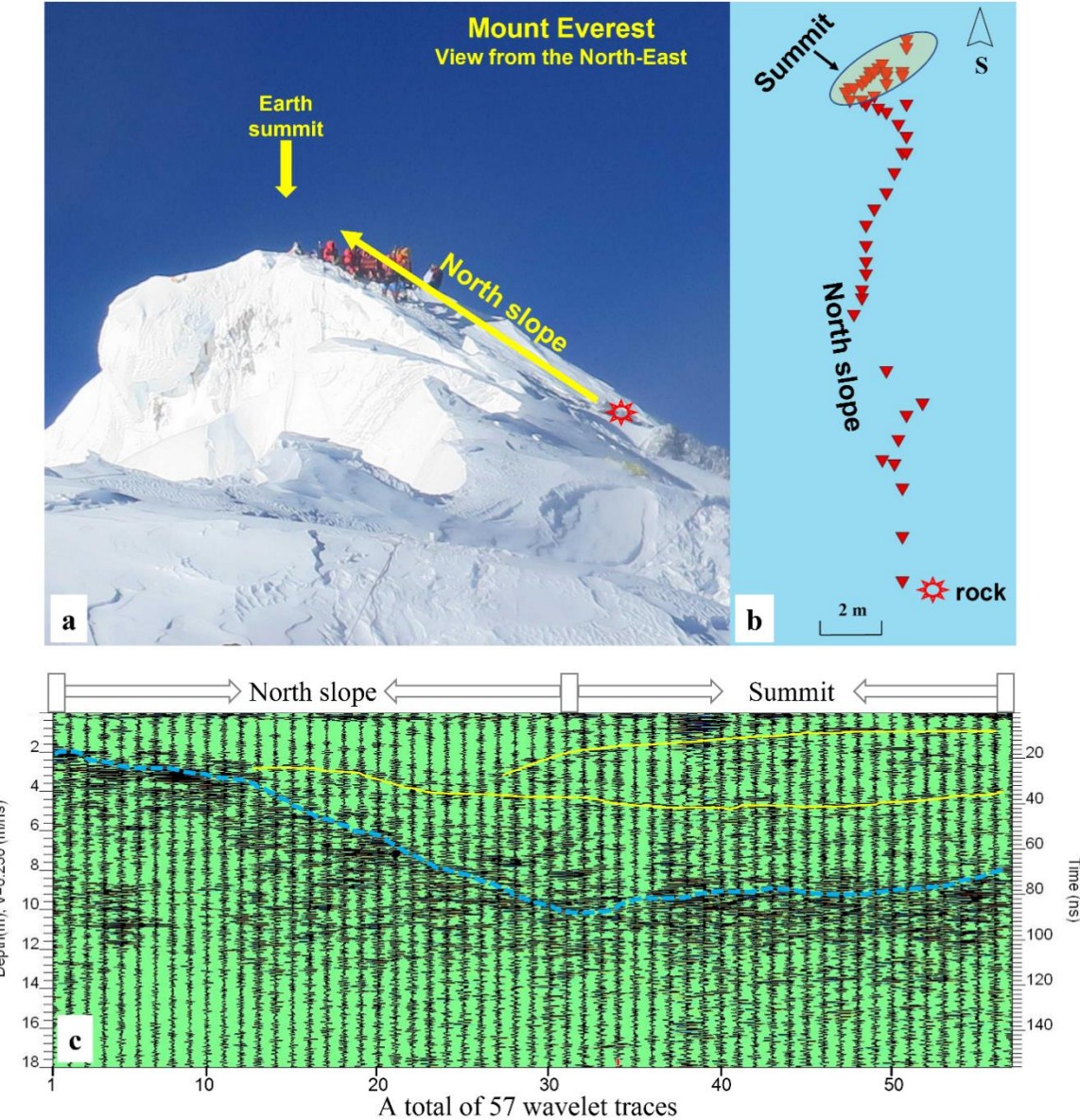

**Figure 1.** Radar measurements along the north slope to Mount Everest acquired on May 4th, 2022. (a) Photo of Mount Everest showing the summit topography in 2022 and the radar measurement direction, as viewed from the northeast. (b) Distribution of 57 radar measurement points (red triangles), which started at the downwards-exposed metamorphosed limestone. (c) Radar wavelet traces showing the boundary between the snow and rock (blue dashed line) and the possible internal stratigraphies (yellow dashed lines) along the radar measurement profile at the estimated depth according to a constant transmission velocity (left axis) and the two-way wave travel time (right axis).

