# Peer review of "Brief communication: How deep is the snow on Mount Everest?"

_The Cryosphere, 2022_

## Author Comment (AC1)

**Referee #1 Response Letter – Manuscript tc-2022-268**

**General Comment:** The authors present a unique dataset - a geophysical survey acquired on the north ridge of Mount Everest, that shows snow depth with distance from the peak. It provides more detailed observations than are otherwise available, and suggests overall that the snow/firn is deeper than previously thought. I am not entirely convinced that these measurements are critical for studies of the cryosphere and climate change (line 25), but they are certainly of broad interest and provide a small window into one of the most inaccessible places on our planet, and for that main reason I'd be pleased to see them published. However, before they are, I would suggest some revisions are required in the way the manuscript is presented:

Reply: We would like to thank the anonymous reviewer for her/his helpful comments on our manuscript, and would hereby like to address the concerns your raised. We think that snow layering at mountain summits contains information about local seasonal snow accumulation and climate history. The comparisons of snow depth/stratigraphy during different time periods may be potentially helpful for understanding the possible influence of anthropogenic climate change at the extreme high elevations in the Himalayas. We will reorganize the abstract and first paragraph of introduction. The first part of introduction will be revised as the following:

"Mount Everest, one of inaccessible places on our planet, is considered to be the most iconic peak (Kang et al., 2022; Matthews et al., 2020). There are very strong scientific and public motivations for determining the snow depth at Mount Everest. Although China and Nepal jointly declared that the snow height of Mount Everest is 8848.86 metres above sea level (m asl) in 2020, the true rock height was not precisely determined due to the unknown snow depth below. The snow depth at extreme high elevations may vary dynamically with different seasons and years. Knowledge about snow depth during different periods will be helpful for explaining the discrepancy of reported snow heights at Mount Everest, which has been introduced by repeated surveys (Angus-Leppan, 1982; Chen et al., 2010; Xie et al., 2021). In addition, snow layering at mountain summits contains information about local seasonal snow accumulation and climate history. Snow and ice displayed an accelerated losing rate in almost all regions on Earth(Hugonnet et al., 2021; Kraaijenbrink et al., 2021). Similar to other snow/glacier-covered summits (Thompson et al., 2009), the snow and glaciers at Mount Everest are the sentinels for climate change and therefore offer a potential natural platform for understanding ongoing climate change at such extreme high elevations (Matthews et al., 2020; Potocki et al., 2022) and their possible widespread influence on the Asian Water Tower (Immerzeel et al., 2020). The comparisons of snow depth/stratigraphy during different periods may be potentially helpful for understanding the possible influence of anthropogenic climate change at the extreme high elevations in the Himalayas (Brun et al., 2022; Pepin et al., 2022; Potocki et al., 2022)."

**Specific Comment:** - the paragraph starting line 26 provides a critical assessment of previous attempts to measure snow depth at the summit, but I can't find any suggestion in the cited papers that there were major doubts in the measurement. I can easily imagine that there is great variability in snow depth depending on exactly where you acquire it from, and at what time of the season you take the measurement. I suggest the authors repackage the paragraph as being a summary of previous work rather than those previous attempts not being successful?

Reply: We agree with your comments. In the revised manuscript, we will repackage the paragraph as a summary of previous works. We will point out the inconsistency of the published snow depth and their possible uncertainties by different methods. We propose to clarify this as follows.

"Previously reported snow depths derived by different methods and instruments range from 0.92 m to 3.5 m at Mount Everest. In 1975, a Chinese expedition team reported an estimated snow depth of 0.92 m by inserting a wooden stake into the snow (Chen et al., 2010). In 1992, a China-Italian joint expedition team estimated a thickness of 2.52 m by inserting a steel stake into the snow (Chen et al., 2010). These results derived by stake methods suffered from many factors such as snow density, stake length, and manpower in such harsh altitude. Radio echo sounding is a suitable technique for imaging snow-ice environments and their internal structures (Rignot et al., 2013). In 2005, a Chinese mountaineering and surveying team claimed a snow depth of ~3.5 m by utilizing ground penetrating radar; however, the reported boundary between the snow and rock on the radar image was too ambiguous to provide an undisputed depth (Sun et al., 2006). In 2019 and 2020, various Nepalese and Chinese expedition teams measured the snow depth using different radar instruments; however, no results were reported. Supported by the Second Tibetan Plateau Scientific Expedition and Research, we organized "The Earth Summit Missions 2022" expedition during the period from April to May 2022. One of our key goals is to measure the snow depth at Mount Everest."

- Related to this, I'd be interested to hear the authors view on whether the timing (season) of the survey makes much of a difference to the snow depth at the summit. Might these results differ if acquired in the post-monsoon, or can we consider them to be consistent throughout the year? I think this is needed to put this snapshot into some sort of broader (longer) context - and very pertinent as the authors themselves state the temporal variability is significant (line 18). A couple of lines added to Section 3 would be good to see in this regard.

Reply: Thank you for your suggestion. As you are concerned, the time (season) of the survey affects the snow depth on the summits and the north slope of Mount Everest. During the "Earth Summit Missions 2022" expedition, another important key task is to install an automatic weather station near the summit of Mount Everest. Based on the experience of mountaineers who have successfully climbed Mount Everest several times, we selected a suitable flat rock surface at about 8800 meters above sea level

for installation. However, when we reach this altitude in May 2022. This selected rock surface was completely covered by snow with a depth of 60-70 cm. We need to move the AWS location to the upper zone. Therefore, we assumed that the snow depth at the summit may be variable at different survey seasons. In this brief communication, we have only reported the average snow depth at the summit of Mount Everest in May 2022. The future repeated radar measurements would give the answer whether there will be a big difference (decrease or increase) in different years with climate change. Following your suggestion, the above information on seasonal snow change will be added in section 3.3.

"In fact, the snow depth on Mount Everest should show the inter-annual variability due to the influence of snow accumulation and snow drift. According to the experience of the mountaineers who reached the summit in 2021, the previously exposed rock surface in May 2021 was covered by a snow cover of about 60-70 cm in May 2022. Our reported snow depth on Mount Everest in 2022 is significantly deeper than the previously reported values during the last five decades (0.9~3.5 m). There is still no solid evidence that the snow cover has become thicker or thinner during the past decades. The future repeat radar measurement at the summit would be helpful to prove such dynamic changes under the background of climate change."

- I have never had the priviledge of summitting Mount Everest, but it looks to me from the photograph in Figure 1 that there is exposed bedrock very close to the surface at the summit location. According to the annotation, the survey profile passes almost directly over that exposed bedrock, but there is no evidence of it in the radargram. I'd be keen for the authors to provide some explanation for this.

Reply: Our radar measurement started near the exposed metamorphosed limestone, at an elevation of about 15 m below Mount Everest, to ensure a gradual transition in the radar reflection profile and thus make it easier to distinguish between snow and rock. Since the starting exposed rock, the measured profile is completely covered by snow. Please see the following screenshot of the video down perspective. Therefore, the radargram shows a gradual deepening trend along the north slope.

[Figure]

- The data presented here are along one survey line, chosen (presumably) to coincide with the established climbing route. I would like to see some acknowledgement that moving the profile several metres either side (even though this might not be safe in practice) could yield very different results. The way the data are presented at the moment is as if these measurements represent snow depth across the whole of the north ridge.

Reply: We agree with your concerns. Our radar line coincided with the established climbing route. Due to local topographic influences, the snow depth along the north slope is heterogeneous. As shown in Figure 1a, the snowpack is significantly deeper near the cliff. Therefore, as you stated, the snow depth would be very different if the radar measurement profile was several meters away from the profile. However, the main objective of this study is to provide the snow depth at the summit of Mount Everest. The radar profile from the exposed metamorphosed limestone to the summit is designed to provide a gradual transition in the radar reflectivity profile, making it easier to distinguish between snow and rock. The high concentration of radar measurements at the summit (No. 32-57) is intended to obtain the mean depth of the snowpack. The future repeat measurement at the summit would therefore be comparable to our measurements without taking into account the geographical differences. Following your suggestions, the relevant explanatory text will be added in the revised manuscript.

"It should be noted that such a measurement along the north slope was used only for the purpose of generating the post-discerning radar boundaries, and the measurement could give different results if the measurement profile were moved a few meters to either side."

- Related to that, since the exact location of the geophyical survey is critical to the data that are retrieved, I'd like to see some precise co-ordinates added to the manuscript (maybe as graticules on Figure 1b) so that anyone wishing to repeat the survey in the future can do so with confidence.

Reply: Thanks for this suggestion. We will provide a table of GPSs in the supplementary materials.

A few additional minor points:

- it would be normal to simply use the term 'Mount Everest' rather than 'the Mount Everest'.

Reply: We will use the term of Mount Everest in the whole text.

- given how critical the transmission velocity is in determining the thickness, I suggest adding an uncertainty range to each of the stated values (based on a min of 0.2 m/ns and a max of 0.27 m/ns).

Reply: Thank you for this suggestion. In the revised manuscript, we will report the uncertainty of the snow depth by applying the range of ±0.03 m/ns (the lower limit of 0.2 m/ns and the upper limit of 0.26 m/ns). Therefore, the mean snow depth at the summit of Mt. Everest in May 2022 was estimated to be 9.5 ± 1.2 m.

- I struggle to make out the red line on Figure 1c - experiment with some different (lighter?) colours?

Reply: For more clarify, we will change the red color to lighter blue colors in the revised manuscript.

---

## Author Comment (AC2)

**Referee #2 Response Letter – Manuscript tc-2022-268**

**General Comment:** TC-2022-268 is a timely and important contribution to quantify the snow depth on the north side of the Mount Everest summit. I congratulate the authors and the entire expedition team on a herculean effort to obtain such valuable data in exceptionally challenging conditions. Although the manuscript represents an important contribution to the cryosphere and broader geophysical communities, there are a number of issues referenced below that need to be addressed prior to eventual publication.

Reply: Thank you for your positive appreciation of our work and the constructive comments, which will help us to improve the paper considerably.

**Specific Comment**

1.   More background is needed in the Introduction about the significance of summit snow depth in the context of climate variability and change. The geodesy discussion is relevant but perhaps a bit tangential to the climate change and cryosphere connection. Also, additional discussion about why/how the summit snow depth is an important indicator of the cryosphere response to climate change would be helpful. This could be done in the context of the Potocki et al. (2022) and Brun et al. (2022) articles which are already cited. I also recommend limiting citations to peer-reviewed scientific articles as much as possible.

Reply: Thank you for your suggestion. Some sentences will be added to address that snow layering at mountain summits contains information about local seasonal snow accumulation and climate history. Knowledge of snow depths is required to estimate the snow water equivalent. The comparisons of snow depth/stratigraphy during different time periods may be potentially helpful for understanding the possible influence of anthropogenic climate change at the extreme high elevations in the Himalayas. We will reorganize the abstract and first paragraph of introduction. The first part of introduction will be revised.

Following your suggestion, we will limit the non-peer-reviewed citations. And some new peer-reviewed scientific articles will be referenced in the revised manuscript.

1.   Pepin, N., Adler, C., Kotlarksi, S., and Palazzi, E.: Mountains undergo enhanced impacts of climate change, Eos Earth & Space Science News, 103, 2022.
2.   Hugonnet, R., McNabb, R., Berthier, E., Menounos, B., Nuth, C., Girod, L., Farinotti, D., Huss, M., Dussaillant, I., and Brun, F.: Accelerated global glacier mass loss in the early twenty-first century, Nature, 592, 726-731, 2021.
3.   Kraaijenbrink, P. D. A., Stigter, E. E., Yao, T., and Immerzeel, W. W.: Climate change decisive for Asia's snow meltwater supply, Nature Climate Change, 11, 591-597, 2021.

The relevant sentence in the background will be changed as following:

"In addition, snow layering at mountain summits contains information about local seasonal snow accumulation and climate history. Snow and ice displayed an accelerated losing rate in almost all regions on Earth (Hugonnet et al., 2021; Kraaijenbrink et al., 2021). Similar to other snow/glacier-covered summits (Thompson et al., 2009), the snow and glaciers at Mount Everest are the sentinels for climate change and therefore offer a potential natural platform for understanding ongoing climate change at such extreme high elevations (Matthews et al., 2020; Potocki et al., 2022) and their possible widespread influence on the Asian Water Tower (Immerzeel et al., 2020). The comparisons of snow depth/stratigraphy during different periods may be potentially helpful for understanding the possible influence of anthropogenic climate change at the extreme high elevations in the Himalayas (Brun et al., 2022; Pepin et al., 2022; Potocki et al., 2022)."

2. A bit more discussion about the hypothesized physical processes responsible for the seasonal and inter-annual variability of summit snow depth could be helpful. Do the authors suggest that the snow accumulation is the result of snowfall (precipitation) or primarily deposition (snow drift) from snow blowing up from lower slopes? How much ablation can be expected due to sublimation? Is there any evidence of melt in the GPR data, which both Matthews et al. (2020) and Potocki et al. (2022) suggest may now occur even at the summit?

Reply: Sorry that we have no evidence to quantify the relative contribution from snowfall (precipitation) and the snow deposition from snow blowing up from slopes. According to the climbers who reached the summit in both 2021 and 2022, the fresh snowpack near the summit is much deeper in 2022 than in 2021. We intend to install an automatic weather station on the rock surface at ~8800m, where there is free snow in May 2021. However, this site was completely covered by a snowpack of 60-70 cm in May 2022. Therefore, we believe that the snow depth at the summit of Mt. Everest should show the seasonal and inter-annual variability. In the section 3 of the revised manuscript, we will add this information to address the temporal changes in snow depth as follow.

"In fact, the snow depth on Mount Everest should show the inter-annual variability due to the influence of snow accumulation and snow drift. According to the experience of the mountaineers who reached the summit in 2021, the previously exposed rock surface in May 2021 was covered by a snow cover of about 60-70 cm in May 2022. Our reported snow depth on Mount Everest in 2022 is significantly deeper than the previously reported values during the last five decades (0.9~3.5 m). There is still no solid evidence that the snow cover has become thicker or thinner during the past decades. The future repeat radar measurement at the summit would be helpful to prove such dynamic changes under the background of climate change."

Regarding the questions of sublimation and melting at the summit, we are sorry that our radar measurement cannot give the answer. In fact, we are also interested in the melting and sublimation at the summit. Therefore, during "The Earth Summit

Missions 2022" expedition in May 2022, we installed an automatic weather station near the summit. Based on this meteorological data, we hope to get the final answer. However, it is beyond the scope of this manuscript. Sorry that we will not include the relevant information and discussion in the revised manuscript.

3. Do the GPR data provide any indication of whether some of the summit snow depth could be the result of rime ice accretion during the monsoon, similar to what occurs in Patagonia? See: Whiteman, C. D., and R. Garibotti, 2013: Rime Mushrooms on Mountains: Description, Formation, and Impacts on Mountaineering. Bull. Amer. Meteor. Soc., **94**, 1319–1327, https://doi.org/10.1175/BAMS-D-12-00167.1.

Reply: Thanks for providing this paper. We are sorry that the GRP data did not provide such information. However, we believe that the mechanism of snow accumulation at the summit of Mount Everest is different from that of rime ice, partly due to the extreme high altitude and contrasting climatic background. As shown in Figure 1a, the summit is covered by snowpack rather than rime ice. Furthermore, as stated by Whiteman and Garibotti (2013), "*The highest summits in the Himalayas have a more continental climate and are more likely to build up snow cornices downwind of obstacles rather than rime mushrooms on the upwind side*'.

4. Is a different photo available for Figure 1a? Are the darker colors below the summit old prayer flags, lower albedo snow, or rock? It is very hard to tell in this photo but I suppose old prayer flags? An improved photo could help this interpretation as the darker colors could easily be mistaken for rock?

Reply: Thanks for pointing that out. Yes, the darker colors below the summit are the old prayer flags. In the revised manuscript, we will update a new photo taken in May 2022 from the similar perspective.

[Figure]

5. The authors reference the importance of future snow core drilling and repeated GPR measurements. Are there any lessons learned from the 2022 expedition and/or suggestions for future expeditions/researchers? Additionally, are you able to offer testable hypotheses for future researchers?

Reply: Future synchronous snow coring and radar measurements at the summit would be valuable. The snow cores are useful to validate the radar measurements and thus to calibrate the snow transmission velocity. And more importantly, given your concerns about surface melting, the analysis of snow stratigraphy would provide important information by verifying the possible melt refreezing layers in the snow core. Such work, together with the ongoing AWS measurement near the summit, would possibly determine the influence of anthropogenic climate change on the Earth's summit. And the repeated radar measurements (perhaps several years later) would be helpful to understand the changes in snow dynamics at this extreme high altitude and to determine whether the snow height of Mount Everest (8848.86 meters in 2020) will change significantly in the future. Based on the dGPS measurement, some interesting scientific and public questions, such as whether the height of Mount Everest is

increasing/decreasing, could be answered. Finally, we will briefly add some perspectives in the revised manuscript.

6.  I suggest using either snow thickness, snow depth, or snow height and being consistent instead of using multiple terms to refer to the same thing which may confuse the reader. Snow depth is perhaps a more commonly used term?

Reply: We will use the consistent term "snow depth" in the revised manuscript.

7.  I suggest consistent use of Mount Everest vs. Mount Chomolungma throughout.

Reply: We will use the consistent term Mount Everest in the revised manuscript.

8.  Are there GPS height measurements for the rock indicated by the blue star in Figures 1a and 1b?

Reply: The portable GPS has large uncertainties in the vertical direction. As shown in Figure 1, the rock is about 15 m lower than the summits. Sorry that we will not added such information in the revised manuscript.

9.  Can contour lines and the international border be added to the map in Figure 1b?

Reply: It is a pity that there are no high resolution DEMs available to create accurate contour lines. We have tried several DEMs including SRTM DEM and High Asia DEM. But the performance is very poor. Therefore, in the revised manuscript, unfortunately, we will not add this information. Perhaps future unmanned aerial vehicle (UAV) survey by structure-from-motion/radar could possibly provide high-resolution DEM for making the contour line at the Earth's summit.

**Minor Comments**

Line 7: "the" preceding Mount Everest not needed here or elsewhere

Reply: We will change it.

Line 17: citation needed for China and Nepal height declaration

Reply: China and Nepal jointly declared that the snow height of Mount Everest is 8848.86 metres above sea level (m asl) in 2020 [ http://www.xinhuanet.com/english/2020-12/08/c_139573400.htm ]. However, no peer-reviewed scientific article on the snow height of Mount Everest of 8848.86 metres is published. Therefore, we did not add the website linage in the revised manuscript.

Line 18: considerable inter-annual variability in the snow thickness may also exist?

Reply: As above reply, we think that there is inter-annual variability on Mount Everest. However, the magnitude of such variability should be quantified by the next repeated radar measurements or core drilling at the summit.

Line 21: suggest changing "In additions," to "In addition,"

Reply: We will change it.

Line 23: suggest changing "extreme high elevation" to "extreme high elevations"

Reply: We will change it.

Line 24: suggest changing "the state of snow at the Mount Everest are critical" to "snow depth at the Mount Everest summit is critical"

Reply: We will change it.

Line 49: remove "t" after "Mount Everest t."

Reply: We will change it.

Lines 53-54: What were the snow properties at 6500 m and 7028 m in 2005 and how certain are you that these properties are representative of the summit snow in 2022?

Reply: In general, the transmission velocity in snow ranges from 0.20 m/ns to 0.27 m/ns, which depend on snow properties (Fortin and Fortier, 2001; Singh et al., 2017). It is pity that we did not measure common midpoint data to evaluate the transmission velocity of radar waves inside the snowpack at the Mount Everest because of the limited measurement time window in so-called 'death zone'. The snowpack ridge of 7028 m asl was deposited by both snow fall and snow drift, which is similar to the summit. The previous measured transmission velocity of 0.23 m/ns was adopted in this study. As pointed by both reviewers, the possible uncertainties of transmission velocity should be addressed in the revised manuscript. Therefore, we will provide the uncertainty of snow depth by applying the range of ±0.03m/ns (the low boundary of 0.2m/ns and upper boundary of 0.26 m/ns). Therefore, the mean snow depth at the summit of Mount Everest was estimated to be 9.5±1.2m in May 2022. And the two-way wave travel time of radar also provided for future comparison by repeated radar measurements.

Line 55: suggest changing "processing package by apply a frequency" to "processing package by applying a frequency"

Reply: We will change it.

Line 69: space needed after "velocity."

Reply: We will change it.

Line 81: suggest changing "was compacted for producing high snow density." to "was compacted resulting in high snow density."

Reply: We will change it.

Line 82: suggest changing snow density to kg m-3

Reply: We will change it.

Line 85: suggest changing "In addition to reveal the" to "In addition to revealing the"

Reply: We will change it.

Line 87: suggest deleting "was" between "reflection layer was existed"

Reply: We will change it.

Line 88: suggest changing "maybe" to "may be"

Reply: We will change it.

Lines 92-93: suggest adding "the" between "of snowpack" and deleting "in the world"

Reply: We will change it.

Lines 96-97: incomplete sentence starting "It is worth noting . . ." and therefore suggest revising

Reply: We will change it.

Line 98: suggest changing "at the Mount Everest is also necessary" to "at Mount Everest are also necessary"

Reply: We will change it.

Line 99: suggest deleting "favor" and "source"

Reply: We will change it.

---

## Author Response (AR1)

**General Comment:** The authors present a unique dataset - a geophysical survey acquired on the north ridge of Mount Everest, that shows snow depth with distance from the peak. It provides more detailed observations than are otherwise available, and suggests overall that the snow/firn is deeper than previously thought. I am not entirely convinced that these measurements are critical for studies of the cryosphere and climate change (line 25), but they are certainly of broad interest and provide a small window into one of the most inaccessible places on our planet, and for that main reason I'd be pleased to see them published. However, before they are, I would suggest some revisions are required in the way the manuscript is presented:

Reply: We would like to thank the anonymous reviewer for her/his helpful comments on our manuscript, and would hereby like to address the concerns your raised. We think that snow layering at mountain summits contains information about local seasonal snow accumulation and climate history. The comparisons of snow depth/stratigraphy during different time periods may be potentially helpful for understanding the possible influence of anthropogenic climate change at the extreme high elevations in the Himalayas. We have reorganized the abstract and first paragraph of introduction. The first part of introduction will be revised as the following:

"Mount Everest, one of the most inaccessible places on our planet, is considered to be the most iconic peak (Kang et al., 2022; Matthews et al., 2020). There are very strong scientific and public motivations for determining the snow depth at Mount Everest. Although China and Nepal jointly declared that the snow height of Mount Everest was 8848.86 metres above sea level (m asl) in 2020, the true rock height has not been precisely determined due to the unknown snow depth below. The snow depths at extremely high elevations may vary dynamically with different seasons and years. Knowledge about snow depths during different periods will be helpful for explaining the discrepancies in the reported snow heights at Mount Everest, which have been introduced by repeated surveys (Angus-Leppan, 1982; Chen et al., 2010; Xie et al., 2021). In addition, snow layering at mountain summits contains information about local seasonal snow accumulation and climate history. However, snow and ice display accelerated loss rates in almost all regions on Earth (Hugonnet et al., 2021; Kraaijenbrink et al., 2021). Similar to other snow/glacier-covered summits (Thompson et al., 2009), the snow and glaciers at Mount Everest are the sentinels for climate change and therefore offer a potential natural platform for understanding ongoing climate change at such extremely high elevations (Matthews et al., 2020; Potocki et al., 2022) and their possible widespread influence on the Asian Water Tower (Immerzeel et al., 2020). Comparisons of snow depth/stratigraphy during different periods may be potentially helpful for understanding the possible influence of anthropogenic climate change at extremely high elevations in the Himalayas (Brun et al., 2022; Pepin et al., 2022; Potocki et al., 2022). "

**Specific Comment:** - the paragraph starting line 26 provides a critical assessment of previous attempts to measure snow depth at the summit, but I can't find any suggestion in the cited

papers that there were major doubts in the measurement. I can easily imagine that there is great variability in snow depth depending on exactly where you acquire it from, and at what time of the season you take the measurement. I suggest the authors repackage the paragraph as being a summary of previous work rather than those previous attempts not being successful?

Reply: We agreed with your comments. In the revised manuscript, we repackaged the paragraph as a summary of previous works. We pointed out the inconsistency of the published snow depth and their possible uncertainties by different methods. We proposed to clarify this as follows.

"Previously reported snow depths derived by different methods and instruments ranged from 0.92 m to 3.5 m at Mount Everest. In 1975, a Chinese expedition team reported an estimated snow depth of 0.92 m by inserting a wooden stake into the snow (Chen et al., 2010). In 1992, a joint Chinese-Italian   expedition team estimated a thickness of 2.52 m by inserting a steel stake into the snow (Chen et al., 2010). These results derived by stake methods were subjected to many factors such as snow density, stake length, and manpower issues at such harsh altitudes. Radio echo sounding is a suitable technique for imaging snow-ice environments and their internal structures (Rignot et al., 2013). In 2005, a Chinese mountaineering and surveying team claimed a snow depth of ~3.5 m by utilizing ground penetrating radar; however, the reported boundary between the snow and rock on the radar image was too ambiguous to provide an undisputed depth (Sun et al., 2006). In 2019 and 2020, various Nepalese and Chinese expedition teams measured the snow depth using different radar instruments; however, no results were reported. Supported by the Second Tibetan Plateau Scientific Expedition and Research, we organized "The Earth Summit Missions 2022" expedition during the period from April to May 2022. One of our key goals was to measure the snow depth at Mount Everest."

- Related to this, I'd be interested to hear the authors view on whether the timing (season) of the survey makes much of a difference to the snow depth at the summit. Might these results differ if acquired in the post-monsoon, or can we consider them to be consistent throughout the year? I think this is needed to put this snapshot into some sort of broader (longer) context - and very pertinent as the authors themselves state the temporal variability is significant (line 18). A couple of lines added to Section 3 would be good to see in this regard.

Reply: Thank you for your suggestion. As you are concerned, the time (season) of the survey affects the snow depth on the summits and the north slope of Mount Everest. During the "Earth Summit Missions 2022" expedition, another important key task is to install an automatic weather station near the summit of Mount Everest. Based on the experience of mountaineers who have successfully climbed Mount Everest several times, we selected a suitable flat rock surface at about 8800 meters above sea level for installation. However, when we reach this altitude in May 2022. This selected rock surface was completely covered by snow with a depth of 60-70 cm. We need to move the AWS location to the upper zone. Therefore, we assumed that the snow depth at the summit may be variable at different survey seasons. In this brief communication, we have only reported the average snow depth at the

summit of Mount Everest in May 2022. The future repeated radar measurements would give the answer whether there will be a big difference (decrease or increase) in different years with climate change. Following your suggestion, the above information on seasonal snow change has been added in section 3.3.

"In fact, the snow depth at Mount Everest should display interannual variability because of the influences of snow accumulation and snow drift. According to the recall of mountaineers who reached the summit in 2021 and 2022, the previously exposed rock surface in May 2021 was covered by snowpack of approximately 60-70 cm in May 2022. Our reported snow depth for Mount Everest in 2022 is considerably deeper than the values that were previously reported during the past five decades (0.9~3.5 m). There is still a lack of evidence that the snowpack has become thicker or thinner in recent decades. Future repeated radar measurements at the summit would be helpful for evidencing such dynamic changes under climate change."

- I have never had the priviledge of summitting Mount Everest, but it looks to me from the photograph in Figure 1 that there is exposed bedrock very close to the surface at the summit location. According to the annotation, the survey profile passes almost directly over that exposed bedrock, but there is no evidence of it in the radargram. I'd be keen for the authors to provide some explanation for this.

Reply: Our radar measurement started near the exposed metamorphosed limestone, at an elevation of about 15 m below Mount Everest, to ensure a gradual transition in the radar reflection profile and thus make it easier to distinguish between snow and rock. Since the starting exposed rock, the measured profile is completely covered by snow. Please see the following screenshot of the video down perspective. Therefore, the radargram shows a gradual deepening trend along the north slope.

[Figure]

- The data presented here are along one survey line, chosen (presumably) to coincide with the established climbing route. I would like to see some acknowledgement that moving the profile several metres either side (even though this might not be safe in practice) could yield very different results. The way the data are presented at the moment is as if these

measurements represent snow depth across the whole of the north ridge.

Reply: We agree with your concerns. Our radar line coincided with the established climbing route. Due to local topographic influences, the snow depth along the north slope is heterogeneous. As shown in Figure 1a, the snowpack is significantly deeper near the cliff. Therefore, as you stated, the snow depth would be very different if the radar measurement profile was several meters away from the profile. However, the main objective of this study is to provide the snow depth at the summit of Mount Everest. The radar profile from the exposed metamorphosed limestone to the summit is designed to provide a gradual transition in the radar reflectivity profile, making it easier to distinguish between snow and rock. The high concentration of radar measurements at the summit (No. 32-57) is intended to obtain the mean depth of the snowpack. The future repeat measurement at the summit would therefore be comparable to our measurements without taking into account the geographical differences. Following your suggestions, the relevant explanatory text has been added in the revised manuscript.

"It should be noted that such a measurement along the north slope was used only for the purpose of generating the post-discerning radar boundaries, and the measurement process could give different results if the measurement profile were moved a few metres to either side. "

- Related to that, since the exact location of the geophyical survey is critical to the data that are retrieved, I'd like to see some precise co-ordinates added to the manuscript (maybe as graticules on Figure 1b) so that anyone wishing to repeat the survey in the future can do so with confidence.

Reply: Thanks for this suggestion. We provided a table of GPSs in the supplementary materials.

A few additional minor points:

- it would be normal to simply use the term 'Mount Everest' rather than 'the Mount Everest'.

Reply: We unified the term of Mount Everest in the whole text.

- given how critical the transmission velocity is in determining the thickness, I suggest adding an uncertainty range to each of the stated values (based on a min of 0.2 m/ns and a max of 0.27 m/ns).

Reply: Thank you for this suggestion. In the revised manuscript, we reported the uncertainty of the snow depth by applying the range of ±0.03 m/ns (the lower limit of 0.2 m/ns and the upper limit of 0.26 m/ns). Therefore, the mean snow depth at the summit of Mt. Everest in May 2022 was estimated to be $9.5 \pm 1.2$ m.

- I struggle to make out the red line on Figure 1c - experiment with some different (lighter?) colours?

Reply: For more clarify, we changed the red color to lighter blue colors in the revised manuscript.

[Figure]

A total of 57 wavelet traces

**Referee #2 Response Letter – Manuscript tc-2022-268**

**General Comment:** TC-2022-268 is a timely and important contribution to quantify the snow depth on the north side of the Mount Everest summit. I congratulate the authors and the entire expedition team on a herculean effort to obtain such valuable data in exceptionally challenging conditions. Although the manuscript represents an important contribution to the cryosphere and broader geophysical communities, there are a number of issues referenced below that need to be addressed prior to eventual publication.

Reply: Thank you for your positive appreciation of our work and the constructive comments, which help us to improve the paper considerably.

**Specific Comment**

1.   More background is needed in the Introduction about the significance of summit snow depth in the context of climate variability and change. The geodesy discussion is relevant but perhaps a bit tangential to the climate change and cryosphere connection. Also, additional discussion about why/how the summit snow depth is an important indicator of the cryosphere response to climate change would be helpful. This could be done in the context of the Potocki et al. (2022) and Brun et al. (2022) articles which are already cited. I also recommend limiting citations to peer-reviewed scientific articles as much as possible.

Reply: Thank you for your suggestion. Some sentences was added to address that snow layering at mountain summits contains information about local seasonal snow accumulation and climate history. Knowledge of snow depths is required to estimate the snow water equivalent. The comparisons of snow depth/stratigraphy during different time periods may be potentially helpful for understanding the possible influence of anthropogenic climate change at the extreme high elevations in the Himalayas. We reorganized the abstract and first paragraph of introduction. The first part of introduction was revised.

Following your suggestion, we limit the non-peer-reviewed citations. And some new peer-reviewed scientific articles were referenced in the revised manuscript.

1.   Pepin, N., Adler, C., Kotlarksi, S., and Palazzi, E.: Mountains undergo enhanced impacts of climate change, Eos Earth & Space Science News, 103, 2022.
2.   Hugonnet, R., McNabb, R., Berthier, E., Menounos, B., Nuth, C., Girod, L., Farinotti, D., Huss, M., Dussaillant, I., and Brun, F.: Accelerated global glacier mass loss in the early twenty-first century, Nature, 592, 726-731, 2021.
3.   Kraaijenbrink, P. D. A., Stigter, E. E., Yao, T., and Immerzeel, W. W.: Climate change decisive for Asia's snow meltwater supply, Nature Climate Change, 11, 591-597, 2021.

The relevant sentence in the background was changed as following:

"In addition, snow layering at mountain summits contains information about local seasonal snow accumulation and climate history. However, snow and ice display accelerated loss rates

in almost all regions on Earth (Hugonnet et al., 2021; Kraaijenbrink et al., 2021). Similar to other snow/glacier-covered summits (Thompson et al., 2009), the snow and glaciers at Mount Everest are the sentinels for climate change and therefore offer a potential natural platform for understanding ongoing climate change at such extremely high elevations (Matthews et al., 2020; Potocki et al., 2022) and their possible widespread influence on the Asian Water Tower (Immerzeel et al., 2020). Comparisons of snow depth/stratigraphy during different periods may be potentially helpful for understanding the possible influence of anthropogenic climate change at extremely high elevations in the Himalayas (Brun et al., 2022; Pepin et al., 2022; Potocki et al., 2022). "

2. A bit more discussion about the hypothesized physical processes responsible for the seasonal and inter-annual variability of summit snow depth could be helpful. Do the authors suggest that the snow accumulation is the result of snowfall (precipitation) or primarily deposition (snow drift) from snow blowing up from lower slopes? How much ablation can be expected due to sublimation? Is there any evidence of melt in the GPR data, which both Matthews et al. (2020) and Potocki et al. (2022) suggest may now occur even at the summit?

Reply: Sorry that we have no evidence to quantify the relative contribution from snowfall (precipitation) and the snow deposition from snow blowing up from slopes. According to the climbers who reached the summit in both 2021 and 2022, the fresh snowpack near the summit is much deeper in 2022 than in 2021. We intend to install an automatic weather station on the rock surface at ~8800m, where there is free snow in May 2021. However, this site was completely covered by a snowpack of 60-70 cm in May 2022. Therefore, we believe that the snow depth at the summit of Mt. Everest should show the seasonal and inter-annual variability. In the section 3 of the revised manuscript, we added this information to address the temporal changes in snow depth as follow.

"In fact, the snow depth at Mount Everest should display interannual variability because of the influences of snow accumulation and snow drift. According to the recall of mountaineers who reached the summit in 2021 and 2022, the previously exposed rock surface in May 2021 was covered by snowpack of approximately 60-70 cm in May 2022. Our reported snow depth for Mount Everest in 2022 is considerably deeper than the values that were previously reported during the past five decades (0.9~3.5 m). There is still a lack of evidence that the snowpack has become thicker or thinner in recent decades. Future repeated radar measurements at the summit would be helpful for evidencing such dynamic changes under climate change."

Regarding the questions of sublimation and melting at the summit, we are sorry that our radar measurement cannot give the answer. In fact, we are also interested in the melting and sublimation at the summit. Therefore, during "The Earth Summit Missions 2022" expedition in May 2022, we installed an automatic weather station near the summit. Based on this meteorological data, we hope to get the final answer. However, it is beyond the scope of this manuscript. Sorry that we will not include the relevant information and discussion in the revised manuscript.

3. Do the GPR data provide any indication of whether some of the summit snow depth could be the result of rime ice accretion during the monsoon, similar to what occurs in Patagonia? See: Whiteman, C. D., and R. Garibotti, 2013: Rime Mushrooms on Mountains: Description, Formation, and Impacts on Mountaineering. Bull. Amer. Meteor. Soc., **94**, 1319–1327, https://doi.org/10.1175/BAMS-D-12-00167.1.

Reply: Thanks for providing this paper. We are sorry that the GRP data did not provide such information. However, we believe that the mechanism of snow accumulation at the summit of Mount Everest is different from that of rime ice, partly due to the extreme high altitude and contrasting climatic background. As shown in Figure 1a, the summit is covered by snowpack rather than rime ice. Furthermore, as stated by Whiteman and Garibotti (2013), "*The highest summits in the Himalayas have a more continental climate and are more likely to build up snow cornices downwind of obstacles rather than rime mushrooms on the upwind side*".

4. Is a different photo available for Figure 1a? Are the darker colors below the summit old prayer flags, lower albedo snow, or rock? It is very hard to tell in this photo but I suppose old prayer flags? An improved photo could help this interpretation as the darker colors could easily be mistaken for rock?

Reply: Thanks for pointing that out. Yes, the darker colors below the summit are the old prayer flags. In the revised manuscript, we updated a new photo taken in May 2022 from the similar perspective.

[Figure]

North slope ⇐⇒ Summit ⇐

A total of 57 wavelet traces

5. The authors reference the importance of future snow core drilling and repeated GPR measurements. Are there any lessons learned from the 2022 expedition and/or suggestions for future expeditions/researchers? Additionally, are you able to offer testable hypotheses for future researchers?

Reply: Future synchronous snow coring and radar measurements at the summit would be valuable. The snow cores are useful to validate the radar measurements and thus to calibrate the snow transmission velocity. And more importantly, given your concerns about surface melting, the analysis of snow stratigraphy would provide important information by verifying the possible melt refreezing layers in the snow core. Such work, together with the ongoing AWS measurement near the summit, would possibly determine the influence of anthropogenic climate change on the Earth's summit. And the repeated radar measurements (perhaps several years later) would be helpful to understand the changes in snow dynamics at this extreme high altitude and to determine whether the snow height of Mount Everest (8848.86 meters in 2020) will change significantly in the future. Based on the dGPS measurement, some interesting scientific and public questions, such as whether the height of Mount Everest is

increasing/decreasing, could be answered. Finally, we briefly added some perspectives in the revised manuscript.

"It is worth noting that recent debates took place on the surface melting that occurred at extremely high elevations (above 8000 m asl) on Mount Everest (Brun et al., 2022; Potocki et al., 2022). Indeed, future snow core drilling and repeated ground penetrating radar measurements at Mount Everest are also necessary to not only increase our understanding of dynamic snow changes, but also to detect the possible influence of unprecedented anthropogenic climate change by exploring the snow stratigraphy and snowpack properties at the Earth's summit."

6.    I suggest using either snow thickness, snow depth, or snow height and being consistent instead of using multiple terms to refer to the same thing which may confuse the reader. Snow depth is perhaps a more commonly used term?

Reply: We used the consistent term "snow depth" in the revised manuscript.

7.    I suggest consistent use of Mount Everest vs. Mount Chomolungma throughout.

Reply: We used the consistent term Mount Everest in the revised manuscript.

8.    Are there GPS height measurements for the rock indicated by the blue star in Figures 1a and 1b?

Reply: The portable GPS has large uncertainties in the vertical direction. As shown in Figure 1, the rock is about 15 m lower than the summits. Sorry that we will not added such information in the revised manuscript.

9.    Can contour lines and the international border be added to the map in Figure 1b?

Reply: It is a pity that there are no high resolution DEMs available to create accurate contour lines. We have tried several DEMs including SRTM DEM and High Asia DEM. But the performance is very poor. Therefore, in the revised manuscript, unfortunately, we will not add this information. Perhaps future unmanned aerial vehicle (UAV) survey by structure-from-motion/radar could possibly provide high-resolution DEM for making the contour line at the Earth's summit.

**Minor Comments**

Line 7: "the" preceding Mount Everest not needed here or elsewhere

Reply: We changed it.

Line 17: citation needed for China and Nepal height declaration

Reply: China and Nepal jointly declared that the snow height of Mount Everest is 8848.86 metres above sea level (m asl) in 2020 [ http://www.xinhuanet.com/english/2020-12/08/c_139573400.htm ]. However, no peer-reviewed scientific article on the snow height of Mount Everest of 8848.86 metres is published. Therefore, we did not add the website linage in the revised manuscript.

Line 18: considerable inter-annual variability in the snow thickness may also exist?

Reply: As above reply, we think that there is inter-annual variability on Mount Everest. However, the magnitude of such variability should be quantified by the next repeated radar measurements or core drilling at the summit.

Line 21: suggest changing "In additions," to "In addition,"

Reply: We have changed it.

Line 23: suggest changing "extreme high elevation" to "extreme high elevations"

Reply: We have changed it.

Line 24: suggest changing "the state of snow at the Mount Everest are critical" to "snow depth at the Mount Everest summit is critical"

Reply: We have changed it.

Line 49: remove "t" after "Mount Everest t."

Reply: We have changed it.

Lines 53-54: What were the snow properties at 6500 m and 7028 m in 2005 and how certain are you that these properties are representative of the summit snow in 2022?

Reply: In general, the transmission velocity in snow ranges from 0.20 m/ns to 0.27 m/ns, which depend on snow properties (Fortin and Fortier, 2001; Singh et al., 2017). It is pity that we did not measure common midpoint data to evaluate the transmission velocity of radar waves inside the snowpack at the Mount Everest because of the limited measurement time window in so-called 'death zone'. The snowpack ridge of 7028 m asl was deposited by both snow fall and snow drift, which is similar to the summit. The previous measured transmission velocity of 0.23 m/ns was adopted in this study. As pointed by both reviewers, the possible uncertainties of transmission velocity should be addressed in the revised manuscript. Therefore, we will provide the uncertainty of snow depth by applying the range of ±0.03m/ns (the low boundary of 0.2m/ns and upper boundary of 0.26 m/ns). Therefore, the mean snow depth at the summit of Mount Everest was estimated to be 9.5±1.2m in May 2022. And the two-way wave travel time of radar also provided for future comparison by repeated radar measurements.

"The transmission velocity in snow generally ranges from 0.20 m/ns to 0.27 m/ns (Kovacs et al., 1995; Fortin and Fortier, 2001; Singh et al., 2017). Taking 0.20 m/ns and 0.26 m/ns as the possible lower and upper boundaries for uncertainty estimation, the mean depth estimates at the summit were approximately 9.5 ±1.2 m in May 2022."

Line 55: suggest changing "processing package by apply a frequency" to "processing package by applying a frequency"

Reply: We have changed it.

Line 69: space needed after "velocity."

Reply: We have changed it.

Line 81: suggest changing "was compacted for producing high snow density." to "was compacted resulting in high snow density."

Reply: We have changed it.

Line 82: suggest changing snow density to kg m$^{-3}$

Reply: We have changed it.

Line 85: suggest changing "In addition to reveal the" to "In addition to revealing the"

Reply: We have changed it.

Line 87: suggest deleting "was" between "reflection layer was existed"

Reply: We have changed it.

Line 88: suggest changing "maybe" to "may be"

Reply: We have changed it.

Lines 92-93: suggest adding "the" between "of snowpack" and deleting "in the world"

Reply: We have changed it.

Lines 96-97: incomplete sentence starting "It is worth noting . . ." and therefore suggest revising

Reply: We have changed it.

Line 98: suggest changing "at the Mount Everest is also necessary" to "at Mount Everest are also necessary"

Reply: We have changed it.

Line 99: suggest deleting "favor" and "source"

Reply: We have changed it.

---

## Author Response (AR2)

**Reply to Review's Comments**

Thanks to the authors for thoroughly considering, and addressing, the reviewer comments. The manuscript is now ready for publication in my view. I have some minor technical suggestions that might be taken care of prior to final acceptance:

**Reply:** Many thanks to the chief editor Chris Derksen and two reviewers. We acknowledged your efforts in the manuscript acknowledgements. "We would like to sincerely acknowledge and express our deep appreciation to the editor Chris Derksen and two anonymous reviewers for their thorough review of this work."

1. Probably the title should finally be 'on Mount Everest' rather than 'at Mount Everest'.

**Reply:** Following your suggestion, we have changed to "on Mount Everest" in the title and whole text in the manuscript and the supplementary.

2. Line 12: remove 'approximately', since you now quote the uncertainties

**Reply:** Done

3. Line 14: replace the tilda (~) with a dash (-) (also elsewhere in the manuscript)

**Reply:** Done

4. Line 24: might read better as 'snow and ice recession is accelerating in almost all regions…'

**Reply:** Done

5. Line 64: amend to 'at an elevation of…'

**Reply:** Done

6. Line 92: amend to 'provide' rather than 'provided'

**Reply:** Done

7. Line 94: amend to 'at an elevation of…'

**Reply:** Done

8. Line 95-96: replace 'at Mount Everest' with 'at the summit'

**Reply:** Done

9. Line 104: remove 'approximately', since you now quote the uncertainties

**Reply:** Done

10: Line 107: amend to 'by a snowpack of approximately 60-70 cm depth…'

**Reply:** Done